# Bioinspired Multipurpose Approach to the Sampling Problem †

Anton Tolstikhin

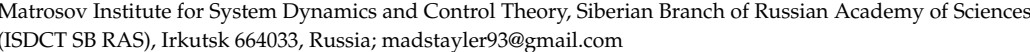

Matrosov Institute for System Dynamics and Control Theory, Siberian Branch of Russian Academy of Sciences (ISDCT SB RAS), Irkutsk 664033, Russia; madstayler93@gmail.com

† The 5th international workshop on information, computation, and control systems for distributed environments.

**Abstract:** Currently, the sampling problem has gained wide popularity in the field of autonomous mobile agent control due to the wide range of practical and fundamental problems described with its framework. This paper considers a combined decentralized control strategy that incorporates both elements of biologically inspired and gradient-based approaches. Its key feature is multitasking, consisting in the possibility of solving several tasks in parallel included in the sampling problem: localization and monitoring of several sources and restoration of the given level line boundaries.

**Keywords:** sampling problem; concentration field; control strategy; autonomous agents; swarm intelligence

## 1. Introduction

The sampling problem [1] is a class of problems that implies a collective study of some given area by a set of coordinated agents. The research objects of the survey are concentration fields—objects, processes, or phenomena that, in general, can be described by some scalar function. At the same time, an important constraint is the ability of agents to interact with the concentration field—to measure the function value—only at their current coordinates and with a given periodicity. Depending on the nature of the field origin, it is commonly accepted to divide them into the following three classes: chemical [2,3], an example of which is the salinity field; physical [4], represented, for example, by thermoclines; and biological [5], which describe the behavior of one or several populations of organisms. Due to such variability and the wide range of both practical and fundamental problems that can be formulated on the basis of the sampling problem, this direction has gained high popularity in the scientific community.

It is usually accepted to distinguish three main purposes of a survey:

- Localization of one or several extrema [6] of the function (sources) describing the concentration field, in the case of a nonstationary field often accompanied by subsequent tracking of their movements in space (monitoring);
- Level lines reconstruction [7], a particular and the most interesting case of which is a search for a zero level line (front) separating an area with positive field values from an area with zero values;
- Mapping [8] of the investigated area, which is often auxiliary for the considered problem and related ones.

Currently, there are a wide range of approaches to solving the sampling problem, which can be roughly divided into the following classes: single-robot control strategies and centralized and decentralized group strategies. The application of each of them is determined by the number of available robots; their hardware characteristics, including computing power and available equipment; the purpose of the survey; and the constraints imposed on this process.

Representatives of the first class are various kinds of trajectory strategies that realize zigzag, tack, spiral, or other maneuvers of agents in space. In this case, the trajectory

can be either predetermined [9,10], departure from which is inadmissible, and adaptively adjusting [11,12] to the current results of problem solving. Obviously, the first option has the lowest requirements for the robot computing power and is the simplest to implement. However, it requires a complete trajectory traversal to announce the results of the survey, has increased requirements for the storage volume of measurement results, and is practically inapplicable in the case of surveying nonstationary concentration fields. The prototypes of the second approach are often bacteria [13,14], insects [15,16], or other biological species [17]. Thus, one of the most well-known strategies is "surge and cast" [18]. It involves straight-line movement towards the estimated location of the field source (surge) and diverging zigzag maneuvers (cast), providing both initial detection of the source trace and its retrieval when the signal is lost. Separately, we should mention the existence of control strategies targeting specific types of robots. As an example, a study [19] applied a gradient-based control strategy for field source search based on the behavior of lobsters. The robot was equipped with several independent sensors that provided an approximation of the field gradient and therefore allowed movement along it.

In general, the simultaneous use of a group of robots can improve both the accuracy and speed of problem solving, which is of particular importance when surveying concentration fields of high size or complex spatial structure. Thus, in the paper [20], a centralized framework is proposed for multiagent control in the task of recovering distributed spatial characteristics of a stationary thermal concentration field. The authors have compared the proposed strategy with a more traditional tack search that results in the demonstration of a significant gain in both speed and accuracy in solving the problem even if the same number of agents is involved. However, such an approach leads to the need for a stable communication channel among the agents, increasing the complexity of the strategy as well as the computational cost. Some of these types of control strategies are developments of ideas from the previous class. For example, in [21], the authors propose an approach in which a concentration field is first surveyed by a group of independent robots moving along predetermined trajectories, after which the collected data are transmitted to a leader machine. The leader, in turn, performs interpolation and extrapolation calculations to reconstruct a stationary concentration field map. Another example is the approach proposed by Petillo [7], which is specially designed for the organization of several vehicles' collective work. According to it, the robots move in parallel courses, performing maneuvers (zigzag and spiral) that are set by the leader. However, each follower, in addition, performs its own zigzag maneuvers along a common course independently of the others. Another type of centralized approaches is paired systems that separate the vehicles that provide the direct solution to the task and those that are third-party observers or computational centers. The latter, in turn, can be either autonomous or manned. As an example, the Fiorelli strategy [22] aims to detect the boundaries of a temperature front by a team of AUVs, supported by a specially equipped aircraft that provides up-to-date measurements of the surveyed field.

On the other hand, decentralized control strategies distribute the computational load evenly among all agents, which positively affects the fault tolerance of the system in general. This class of approaches loads the communication channel to an even greater extent, but this disadvantage is compensated by the lower dependence of the problem solving process on the imposed maximum message transmission range constraints. Thus, in [23], a decentralized adaptive sampling approach based on reinforcement learning during the problem solving process is proposed. The strategy has been compared with the centralized DARP area division algorithm [24], taking into account the different numbers of mobile robots used and the communication range constraints. The results show that the proposed approach performs at least as well as its centralized alternative, and under several test conditions even outperforms it.

Formation-based strategies stand out as they can be implemented either centrally or decentrally. In general, the strategies are reduced to ensuring that the agents hold some given spatial configuration and organize their movement based on a mutual comparison of

the field measurements taken. For example, in [25], an anemotaxis method for controlling V-shaped, linear, and square robot formations is presented. Depending on the goals of the survey, the formation can take different forms—linear, wedge shaped, or circular—each having its positive and negative features [26]. A characteristic drawback of this approach for some types of formations is the lack of data in the center of the group, which can negatively affect the source search subproblem. One of the ways to avoid this issue is an approach proposed by Ogren et al. [27]. It provides interpolation of measurements within the formation based on the least squares method.

With respect to this class of control strategies, the approaches of organization and retention of formation are of most interest. One of the first such approaches, still in use today, is the "follower–leader" approach [28], where one or more vehicles are taken as the leaders and the rest as followers. Leaders are the anchor points around which the formation is built, and also can dictate the reference trajectories for followers. The latter, in turn, track the geometry of the group using distance–distance and distance–angle methods, the combination of which yields various formations, such as triangular, rhombic, and linear [29]. Despite the simplicity of implementation and the scalability of this approach, the main disadvantage is the high dependence of the group on the technical characteristics of the leader, as well as the low robustness with respect to the possibility of its failure.

Another method of formation organization, primarily related to centralized control, is the virtual structure [30,31]. It consists in the creation of virtual points in a single coordinate system for all agents, which determine the mutual location of agents within the group. Each robot tends to minimize the error between its current and desired coordinates. To solve the problem of surveying the concentration field, the entire virtual structure moves as a unit, realizing the approaches discussed above or implementing other solutions unique to this case. Shape control [32] can be considered as a special case, which removes strict requirements on the mutual location of agents and only requires them to be located within the desired subarea (shape).

Finally, the artificial potential function (APF) method, on the contrary, is more often applied in cases of decentralized control of agents. In general, two forces are given: an attractive and a repulsive one, acting on each robot [33]. The first one ensures the unity of the group by preventing agents from leaving the formation and guides them to the desired position, while the second one is responsible for the absence of collisions within the group and obstacle avoidance. The final control is obtained by adding these two forces. The method of artificial potential functions is now widely used in control problems of mobile robots [34–36], including the sampling problem. Its main disadvantage is the possibility of agents getting into the local extrema of the potential function, the exit from which is impossible without the use of additional control mechanisms. The method of behavioral structure proposed by Balch and Arkin [37] can be considered as an ideological development of APF. This approach assumes the simultaneous existence of multiple mission goals (behaviors), each of them influencing the robot in the form of some force. The weighted sum of such forces is applied as control. At the same time, the amplification coefficients can be either predetermined or dynamically changing depending on the process of problem solving and external conditions. As in the method of artificial potential functions, there remains a high complexity of group dynamics formalization, and as a consequence, the stability of such a system cannot be guaranteed. An example of this approach's implementation is the organization of robot swarms' circular motion described in [38].

Based on the overview, it can be concluded that the vast majority of approaches described above only survey stationary and quasi-stationary fields, i.e., those that change at such a small rate that it can be neglected in the framework of the search mission. However, in the case of nonstationary fields, in particular the fields of biological origin, such approaches have a large error in the results. In addition, they often solve only one of the concentration field survey subtasks. In rare exceptions, radically differing algorithms are sequentially applied to each of them. In this paper, we propose a decentralized multiagent control strategy designed to solve in parallel the subtasks of source localization and level

line reconstruction of a nonstationary concentration field. It is based on combining elements of bioinspired [19] and gradient approaches, described in terms of APF.

## 2. Problem Formulation

Let the nonstationary scalar concentration field in the general case be described by the following function:

$$f(t,q) : T \times Q \to \mathbb{R}, \quad Q \subseteq \mathbb{R}^p, \quad T = [0, \infty), \tag{1}$$

where $p \geq 2$.

Four classes of stationary concentration fields are considered in this study, examples of which models are shown in Figure 1. In addition, two models of nonstationary fields of biological nature are presented. This choice is due to the fact that, in most cases, chemical and physical processes are much slower. Therefore, it is possible to neglect their variability within the framework of real problems, considering them as stationary ones, which greatly simplifies the task of the concentration field survey.

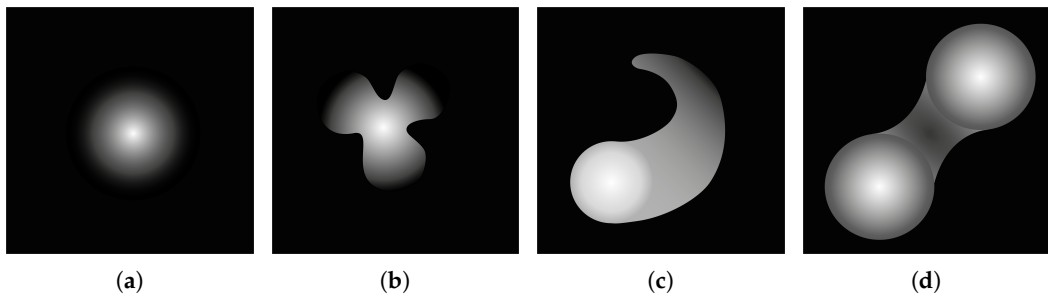

|     |     |     |     |
| --- | --- | --- | --- |
| (**a**) | (**b**) | (**c**) | (**d**) |

**Figure 1.** Simplified cases of stationary concentration field models: (**a**) convex, (**b**) nonconvex (star domain), (**c**) nonconvex (general case), and (**d**) nonconvex with several sources.

The first model describes the interdependent distribution of two continuously growing populations in space and time, which can be represented using the following system that is a special case of [39]:

$$f(t,q) = \left( \phi_1(t) - \frac{1}{r^2} \|q - q_0\|^2 \right)^3,$$

$$g(t,q) = \left( \frac{29}{2048} \right)^3 \left( \phi_2(t) - \frac{1}{r^2} \|q - q_0\|^2 \right)^3, \tag{2}$$

$$t \in [0, \infty), q \in \mathbb{R}^2,$$

where the functions $\phi_i(t)$ describe the growth of agent concentrations at the base boundary with time and are subject to the following system of differential equations:

$$\begin{cases} \dot{\phi}_1(t) = \frac{29}{6} \phi_2(t) - \frac{11}{6} \phi_1(t), \\ \dot{\phi}_2(t) = \frac{4879}{12,288} \phi_1(t) - \frac{4357}{12,288} \phi_2(t), \end{cases} \qquad \phi_i(0) = 1. \tag{3}$$

This model has a number of features:

- Each concentration field has a so-called "base" or source $q^e$, represented by a circle of radius $r$ and centered at $q_0$, the concentration inside which is equal to the concentration on its perimeter. In this case, the source is the same for both populations and cannot move, but the field function values grow exponentially with time.
- Level lines are concentric circles, whose radius growth rate also increases exponentially with time.
- Based on the biological prototype of the model, concentration values cannot take negative values.

The second model describes the swarming movement of one or more independent populations of a fixed size through a closed territory. In this case, the behavior of each individual $q^a$ of population $B$ is modeled according to Reynolds' laws [40], and the field function has the following form:

$$f(t,q) = (a * h)(t,q),$$

$$a(t,q) = \begin{cases} 1 & if \quad \exists k \in B : q = q_k^a \\ 0 \end{cases}, \quad h(t,q) = \begin{cases} 1 & if \quad \|q\| < r \\ 0 \end{cases}, \tag{4}$$

where $(a * h)$ is the convolution operation, and $r$ is the convolution kernel radius, which is a controllable parameter and represents the characteristics of some sensor capable of counting the number of individuals in a given neighborhood.

The key difference of this model is the absence of pronounced field sources. In this case, the source will be understood as a local extremum of the field function, in other words,

$$q_j^e(t) = \{q \in Q : f(t,q) > f(t, q + \epsilon) \quad \forall \|\epsilon\| < \epsilon_0\}, \quad j = 1 \dots n_e, \tag{5}$$

where $\epsilon_0$ is some small neighborhood, and $n_e$ is the number of tracked sources.

The sampling problem in this case consists in a parallel solution of the following subtasks: sources localization and restoration of the given level line. The first one can be divided into detection, localization, and long-term monitoring of all sources of a concentration field. In this case, we assume that $n_e$ is chosen in such a way that the number of available robots is sufficient to fulfill the subtask. Restoration of the level line implies the need for a uniform distribution of agents along the desired level line and tracking its movements in space. What is most interesting in this context is the search for the front — the $0^+$ level line separating the region of positive concentration from monotonic zero values.

To evaluate the current quality of the solution of each subproblem, we introduce the following quality criteria:

$$M_e(t) = \max_{j=(1, n_e)} \min_{i \in G_{base}} \|q_i(t) - q_j^e(t)\|, \tag{6}$$

$$M_f(t) = \sum_{i \in G_{front}} \frac{f(t, q_i(t))}{|G_{front}|} - s, \tag{7}$$

where $q_i(t)$ are the coordinates of the agents used to solve the problem; $G_{base}$ and $G_{front}$ are subsets of agents searching for sources and level lines, respectively; and $s$ is the desired value of the level line. The criterion $M_e(t)$ can only take positive values; accordingly, its minimization demonstrates that each source has at least one agent in its neighborhood. On the other hand, the criterion $M_f(t)$ can take both positive values, indicating that the agents are lagging behind the desired level line, and negative values, signaling its advance. It should be noted that both criteria are used simultaneously, evaluating different aspects of the concentration field survey process: source localization and level line reconstruction, respectively.

## 3. Control Strategy

We assume that the agents used to solve the problem are second-order integrators, and their dynamics are described by the following system:

$$\dot{q}_i = v_i, \quad \dot{v}_i = u_i, \quad i \in G = 1, 2, \dots, n, \quad \|v_i\| \leq v_{max}, \quad \|u_i\| \leq u_{max}, \tag{8}$$

where $q_i \in \mathbb{R}^2$, $v_i \in \mathbb{R}^2$, and $u_i \in \mathbb{R}^2$ are, respectively, the position, speed, and control of the $i$-th agent; $v_{max}$ and $u_{max}$ are the limits of speed and control; and $G$ is the set of available agents with the power of $n$. Each agent is capable of measuring the value of the concentration field at the point of its current location with a given periodicity. In addition,

we assume that each agent can contact any other agent in order to request its current coordinates and the value of the last measurement made.

The proposed strategy specifies the control of each agent as a weighted sum of several forces affecting it:

$$u_i = c_1 F_i^c + c_2 F_i^g + c_3 F_i^s + c_4 F_i^f + c_5 F_i^r + c_6 F_i^b, \tag{9}$$

where $c_1 - c_6$ are some positive coefficients. Thus, to ensure parallelism in solving the subproblems of source localization and level line recovery, the set of agents is divided into two nonoverlapping subsets $G_{base}$ and $G_{front}$ such that $|G_{base}| \ll |G_{front}|$. Depending on the affiliation, the influence of some forces on the agents is eliminated by zeroing out the corresponding coefficients. Thus, for agents providing source search, $c_4$ and $c_5$ are taken to be zero, while for agents of the $G_{front}$ set, this is done with coefficients $c_1 - c_3$.

Since the task of surveying the concentration field implies the need to search and monitor multiple sources in parallel, the $G_{base}$ set is partitioned into several search clusters such that

$$\tau = \{\tau_1, \tau_2, ..., \tau_m\}, \quad \forall j, k : j \neq k \rightarrow \tau_j \cap \tau_k = \varnothing, \quad |\tau_j| \approx |\tau_k| \geq 3.$$

Agents belonging to different clusters are in a competitive relationship with each other. The term "amensalism" is more correct in terms of the biological nature of the proposed control strategy. This means that some agents have a negative effect on others, without experiencing any positive or negative influence from the latter. This is expressed in the fact that the group that is closer to the source repels other clusters, preventing them from approaching.

The gradient force $F_i^g$ (see Figure 2) guides robots along the calculated gradient estimation to the expected extreme field value and is defined as

$$F_i^g = \sum_{j \in \tau_k} \frac{q_{ij}(f(t, q_j(t)) - f(t, q_i(t)))}{\|q_{ij}\|}, \quad i \in \tau_k, \tag{10}$$

where $\|q_{ij}\|$ is the Euclidean norm of the vector $q_{ij} = q_i - q_j$. Due to this force, self-organization of cluster motion is achieved, with each robot moving in the same direction and at the same speed as its neighbors, and an exploitation mechanism is realized.

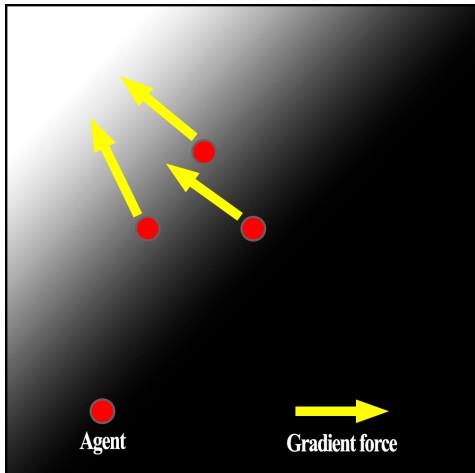

**Figure 2.** Illustration of gradient force application.

On the other hand, the cooperating force $F_i^c$ (Figure 3) ensures the realization of swarm behavior by agents, in particular, avoiding collisions and centering the swarm. It is proposed to define this force as follows:

$$F_i^c = - \sum_{j \in \tau_k} \nabla q_i U_{ij}^c(\|q_{ij}\|), \quad i \in \tau_k, \tag{11}$$

where $U_{ij}^c(\|q_{ij}\|)$ is an artificial potential function that determines the interaction of agents, and $\nabla q_i$ denotes the gradient with respect to the components of the vector $q_i$. The potential function $U_{ij}^c : \mathbb{R}^+ \to \mathbb{R}^+$ is defined as follows:

$$U_{ij}^c(\|q_{ij}\|) = \alpha \left( \frac{1}{2}(\|q_{ij}\| - d_{ij}^A)^2 + \beta \ln \|q_{ij}\| + \beta \frac{d_{ij}^A}{\|q_{ij}\|} \right), \tag{12}$$

where $\alpha, \beta \in \mathbb{R}^+$ are some control parameters, and $d_{ij}^A > 0$ determines the desired distance between agents. The function (12) is based on the potential function proposed in [41], with the addition of a parameter $\beta$ that allows for varying the size of the region where the function is strictly convex. Thus, under the influence of this force, the agents of one cluster tend to form a formation, which in the case of the minimum required number of agents has the form of a regular triangle with faces of length $d_{ij}^A$.

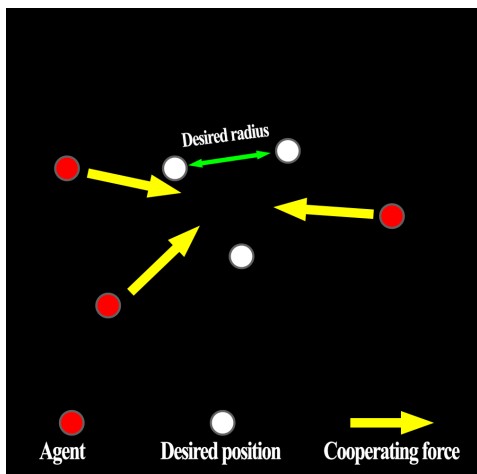

**Figure 3.** Illustration of cooperating force application.

Finally, the segregating force $F_i^s$ is responsible for the above-described interaction between different clusters and is specified as follows:

$$F_i^s = - \sum_{j \notin \tau_k} \nabla q_i U_{ij}^s(\|q_{ij}\|), \quad i \in \tau_k, \tag{13}$$

$$U_{ij}^s(\|q_{ij}\|) = \begin{cases} 0 & if \quad \|q_{ij}\| > d_{ij}^B \vee f(t, q_j(t)) > f(t, q_i(t)) \\ \alpha \left( \frac{1}{2}(\|q_{ij}\| - d_{ij}^B)^2 + \beta \ln \|q_{ij}\| + \beta \frac{d_{ij}^B}{\|q_{ij}\|} \right) & otherwise \end{cases}, \tag{14}$$

where $d_{ij}^B \gg d_{ij}^A$ is the minimum desired distance between clusters. The application of this force is shown in Figure 4. The main difference between Equations (12) and (14) is that the latter does not affect the cluster that is closer to the source, and provides only repulsion beyond the desired radius, rather than maintaining a preferred distance.

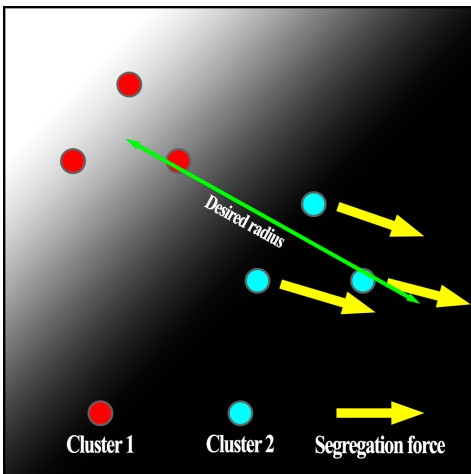

**Figure 4.** Illustration of segregation force application.

Each agent of the $G_{front}$ set is oriented to the position of one of the search clusters $\tau$ when searching for the level line. Initially, this relationship is set with the closest cluster, but the agent can decide to change it under certain conditions. The force $F_i^f$, called frontal because of its original purpose of detecting the front (the $0^+$ level line) of the concentration field, attracts an agent to the current location of the source or repels it (see Figure 5), depending on whether its current measurement value is lower or higher relative to the target level line. In this case, the coordinates of the source are taken as the averaged coordinates of the agents in the bound cluster. In other words,

$$F_i^f = \sum_{j \in \tau^*} \frac{(2g_i(f(t,q_i)) - 1)q_{ij}}{\|q_ij\|}, \tag{15}$$

$$g_i(f(t,q_i)) = \begin{cases} 0 & f(t,q_i) \leq f_l \\ 3\left(\frac{f(t,q_i)-f_l}{f_u-f_l}\right)^2 - 2\left(\frac{f(t,q_i)-f_l}{f_u-f_l}\right)^3 & f_u \geq f(t,q_i) \geq f_l, \\ 1 & f(t,q_i) \geq f_u \end{cases} \tag{16}$$

where $f_l$ and $f_u$ are the lower and upper bounds of the target values, respectively, and $\tau^*$ is the current search cluster associated with the agent. The function $g_i(f(t,q_i))$ is a so-called SmoothStep function that ensures smooth switching between attraction and repulsion modes in the vicinity of the desired level line, due to which the control does not tolerate discontinuities.

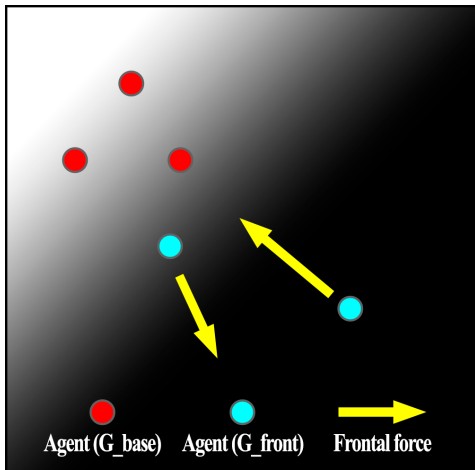

**Figure 5.** Illustration of frontal force application.

The relaxation force $F_i^r$, depicted in Figure 6, is applied to distribute the agents uniformly along the perimeter of the level line and has the following form:

$$F_i^r = \sum_{i,j \in \tau^*} \frac{q_{ij}}{\|q_{ij}\|} \left( e^{d_{ij}^C - \|q_{ij}\|} + a \right), \tag{17}$$

where $d_{ij}^C$ is a desired distance between agents, and $a$ is a positive coefficient determining the rate of agents' distribution along the perimeter. When solving the problem, there may be situations in which too many agents are engaged in the delineation of one level line, while others suffer from agents' shortage. In this case, the following condition will be satisfied:

$$\left( \|c_4 F_i^f\| < \left| \frac{c_4 F_i^f \cdot c_5 F_i^r}{\|c_5 F_i^r\|} \right| \right) \wedge (c_4 F_i^f \cdot c_5 F_i^r < 0), \tag{18}$$

where $\cdot$ is the scalar product operator. If this condition is detected (Figure 7), the agent decides to change the bound cluster to the next one in the looping list. Thus, a self-organization mechanism is implemented, allowing agents of the $G_{front}$ set to migrate from areas with their surplus. In this case, no additional information about the spatial distribution of agents is required. A side effect of the influence of these two forces is that the agents maintain a certain formation when moving through areas with a field function value below the target one.

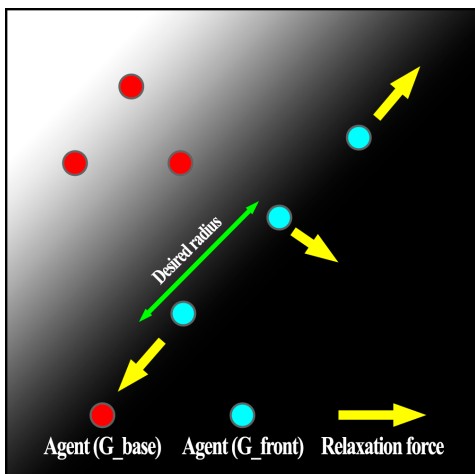

**Figure 6.** Illustration of relaxation force application.

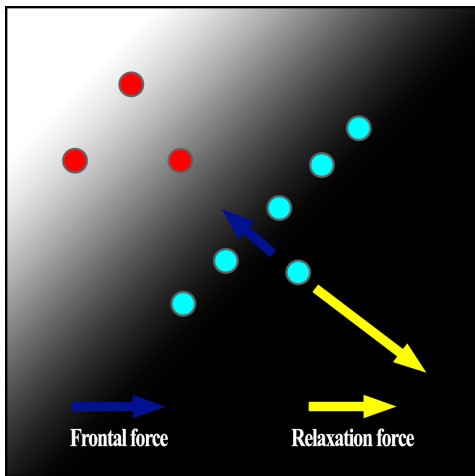

**Figure 7.** Example of the migration condition occurrence.

Finally, the boundary force (Figure 8) is applied to all agents. Its main purpose is to keep agents inside the surveyed area. It is set as follows:

$$F_i^b = [f_1^b, ..., f_p^b],$$
$$f_k^b = e^{x_k^{min} + w - x_k} - e^{x_k - x_k^{max} + w},$$

(19)

where $x_k^{min}$ and $x_k^{max}$ are components of the vectors bounding the surveyed area, $x_k$ are components of the vector $q_i$, and $w$ is the dumping zone size along the perimeter of the surveyed area.

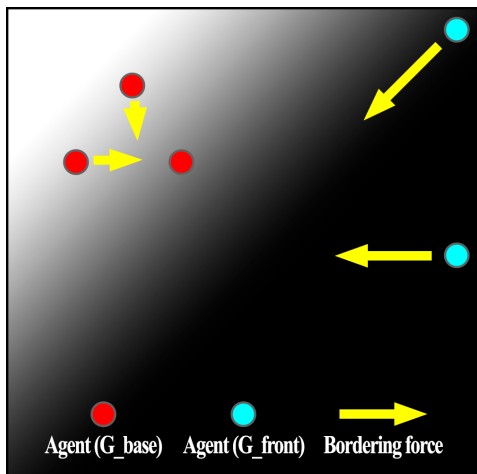

**Figure 8.** Illustration of bordering force application.

## 4. Conclusions

Several series of computer experiments were conducted using the concentration field models described above to evaluate the proposed approach. Figure 9 shows graphs of changes in the efficiency criteria (6) and (7) for the most representative variants of agents' behavior when solving the problem. The conditions of the experiments are given in Table 1.

**Table 1.** Simulation parameters.

| Parameter | Growing Population Model | Swarm Population Model |
|---|---|---|
| Area size | $1000 \times 1000$ m | $1000 \times 1000$ m |
| Simulation time | 3000 model seconds | 3000 model seconds |
| $|\tau|$ | 1 | 3 |
| $|G_{front}|$ | 50 | 50 |
| $v_{max}$ | $3 \frac{m}{s}$ | $3 \frac{m}{s}$ |
| $u_{max}$ | $0.5 \frac{m}{s^2}$ | $0.5 \frac{m}{s^2}$ |
| Target level line | 45 | 45 |

In these graphs, we can conventionally distinguish three segments describing different stages of the problem solution. The first stage is the formation of search clusters and their search for sources. It should be noted that for the first model, this stage takes considerably more time ($t \in [0, 800]$). This is caused by the fact that due to the peculiarities of this model at the early stages, the concentration field function takes zero values practically at all points of the surveyed area. Because of this, the gradient force has no effect on the agents that carry out drift under the influence of the other forces. However, if at least one nonzero measurement is detected, the source is quickly localized.

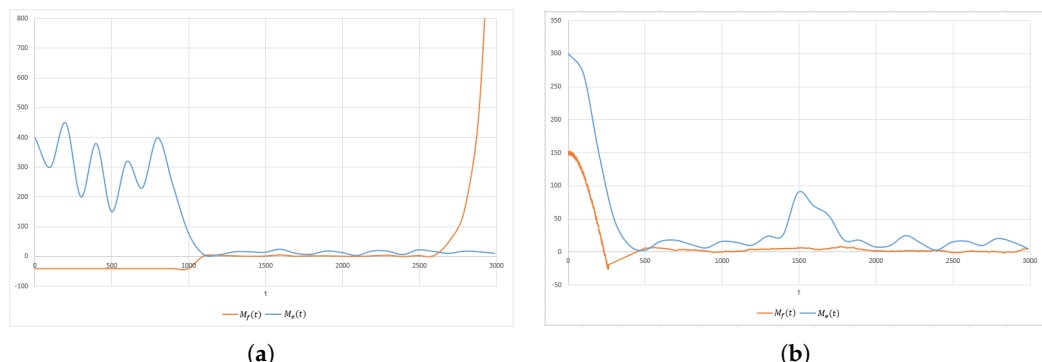

**Figure 9.** Simulation results for the growing population model (**a**) and swarm population model (**b**).

The next stage involves monitoring the sources and bringing the agents of the $G_{front}$ set to the target level line, as well as their uniform distribution along it. Tests have shown that, on average, the deviation of the $M_e(t)$ metric at this stage does not exceed 20 m, which, given the size of the surveyed area and the distance between the agents (25 m), is acceptable. However, due to the hardly predictable movement of sources in the swarm population model, short interruptions of monitoring can be observed.

Finally, the third stage is typical only for the growing population model. Considering the exponential growth of the level line velocity, it is guaranteed that a situation occurs when the agent's velocity constraint makes it impossible to further solve the problem. The last important fact revealed at the stage of testing the strategy is the limitation of the level line classes that can be tracked. This approach can be used if the area bounded by a level line is convex (Figure 10a) or is a star domain (Figure 10b) [42] with respect to the source coordinates. In the case of a nonconvex region (Figure 10c), there may be extended perimeter areas that agents are unable to detect. However, there can potentially be situations (Figure 10d) in which the correct boundaries can be recovered through the collective action of several agent subgroups.

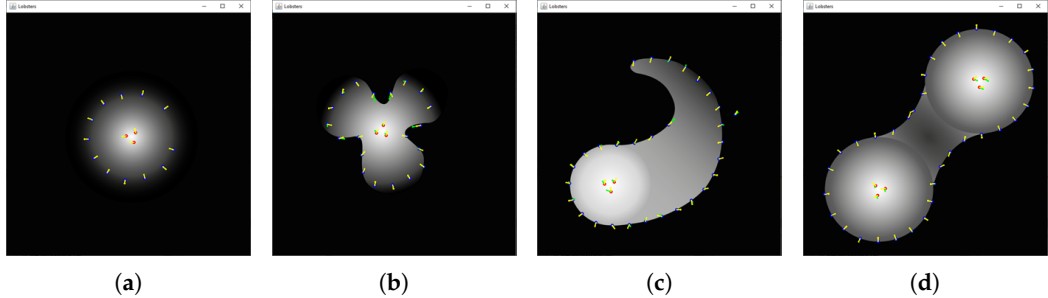

**Figure 10.** Source localization and level line reconstruction for different concentration field types: (**a**) convex, (**b**) nonconvex (star domain), (**c**) nonconvex (general case), and (**d**) nonconvex with several sources.

The results demonstrate the applicability of the strategy for a wide range of concentration field survey problem formulations. However, the testing revealed limitations of the approach, which will be partially or fully eliminated in further stages of the study. The first direction is to develop the ideas of the described self-organization mechanism. As noted earlier, currently, the $G_{front}$ and $G_{base}$ sets, as well as the search clusters $\tau$, are formed before the start of the mission and do not change during its execution. This can negatively impact the survey process if there is a possibility of robot malfunction. In addition, the tests using the swarm population model have shown that there might be rare situations in which the number of simultaneously existing sources exceeds the number of search clusters at a certain time interval. The ability of agents to adaptively switch between clusters should eliminate these disadvantages. The second area of research development

is the development and software implementation of additional nonstationary models of concentration fields with a different nature of origin. In addition, the developed control strategy will be compared with existing sampling problem solutions under uniform and controlled conditions.

**Funding:** This work was funded by the Russian Science Foundation under grant number 22-29-00819.

**Institutional Review Board Statement:** Not applicable.

**Informed Consent Statement:** Not applicable.

**Data Availability Statement:** Data are contained within the article.

**Conflicts of Interest:** The authors declare no conflict of interest.

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
