# Peer review of "Bioinspired Multipurpose Approach to the Sampling Problem†"

_computation, doi:10.3390/computation11120254_

Round 1

Reviewer 1 Report

Comments and Suggestions for Authors

In this paper, the author addresses the problem of sampling in a decentralized mobile agent control application. To do so, he proposes strategies combining bio-inspired and gradient approaches. 

The set of agents is partitioned into non-overlapping subsets to provide parallelism in solving the sub-problems of source localization and level line recovery.

The design of the paper focuses on the evaluation of qualitative criteria for monitoring sources in the agent's neighbourhood. However, it is not clear which of these criteria the author prefers. According to Figure 3, it is clear that in the case of the simulation results for the growing population model the values of the two criteria differ significantly.

The paper lacks validation on a wider set of test problems.

At the same time, the main contribution of the article and comparison with other authors' approaches is not formulated. The analysis in Chapter 1 is very illustrative and readable, but it should also indicate how the author relates to these works and how his approach is better.

It is not clear what is meant by the symbols $\Psi_1(t)$ and $\Psi_2(t)$ on page 4.

In formulas connected by a clamp, in the first case, $\Psi_1(t)$ can be converted from the right side of the equation to the left side, and $\Psi_1(t)$ can then be expressed as a function of $\Psi_2(t)$. Alternatively, for the second equation, $\Psi_2(t)$ can then be expressed as a function of $\Psi_1(t)$.   However, if we compare the values for the same function after such an adjustment, they do not agree.

(This would be relevant when sampling in discrete control theory systems and the time on the left side of the equations was relative to the next instant and the right side was the values at the current instant.)

A few formal comments:

For ease of reference, formulas should be numbered throughout the text (i.e., written in the environment

\begin{equation} \label{eq:...}

...

\end{equation}

There are only two such in the paper on page 5, which define qualitative criteria for monitoring sources in the agent's neighbourhood.

It is unnecessary to mention the same formula twice in the text (on pages 1 and 3), in the descriptive section in the Introduction on page 1, a verbal explanation is sufficient.

p. 2, l.58: "communication channel between the agents" - "… among the agents"?

Fig. 2b): star domain is also non-convex.

Reviewer 2 Report

Comments and Suggestions for Authors

In the reviewer's opinion, the paper needs minor revision.

First, it should be made clear where there are extensions from the original version previously published, as well as the contributions of individual authors.

At the end of the Introduction, the author notes that the paper is based on self-organisation mechanisms. What is missing is a description of swarm self-organisation mechanisms. The force to ensure the realisation of swarm behaviour by agents, in particular collision avoidance and swarm centring, is described.

In the reviewer's opinion, this is not sufficient to write of self-organisation.

Reviewer 3 Report

Comments and Suggestions for Authors

This study explores a novel decentralized control strategy, merging elements from both biologically inspired and gradient-based approaches. While the obtained results present intriguing insights, the current version of the paper falls short of meeting publication standards. In light of this, several constructive suggestions are proposed to enhance the paper:

1.        Enhance the introduction by clearly highlighting the primary contribution of the paper.

2.        Integrate visual aids such as flowcharts to provide a clearer depiction of the control strategy.

3.        Enrich the simulation section with comparative simulations for a more comprehensive analysis.

4.        Revise and update references, ensuring the inclusion of more recent papers published within the last three years.

5.         Thoroughly address grammatical errors and meticulously maintain consistent English tense throughout the manuscript.

Comments on the Quality of English Language

Extensive editing of English language required.

Round 2

Reviewer 1 Report

Comments and Suggestions for Authors

In this paper, the author addresses the problem of sampling in a decentralized mobile agent control application. To do so, he proposes strategies combining bio-inspired and gradient approaches. 

The set of agents is partitioned into non-overlapping subsets to provide parallelism in solving the sub-problems of source localization and level line recovery.

In my first review, I had a number of comments requiring addition, clarification or correction. In his Cover Letter, the author has responded to all of them in detail and satisfactorily, and has edited and expanded the article by 3 pages to highlight his contribution. The combination of two criteria for source localization and level line reconstruction is interesting and original.

Only a response to the comment "The paper lacks validation on a wider set of test problems." referring to future research and " Three more models of chemical and biological nature are currently being developed" is questionable. But it would certainly be interesting to at least present them in a framework. However, it can be agreed that the concept of the paper lies in the design of new models and their basic validation.

Reviewer 3 Report

Comments and Suggestions for Authors

No more comments.